# Coupled Generative Adversarial Networks

**Ming-Yu Liu**
Mitsubishi Electric Research Labs (MERL),
mliu@merl.com

**Oncel Tuzel**
Mitsubishi Electric Research Labs (MERL),
oncel@merl.com

## Abstract

We propose coupled generative adversarial network (CoGAN) for learning a joint distribution of multi-domain images. In contrast to the existing approaches, which require tuples of corresponding images in different domains in the training set, CoGAN can learn a joint distribution without any tuple of corresponding images. It can learn a joint distribution with just samples drawn from the marginal distributions. This is achieved by enforcing a weight-sharing constraint that limits the network capacity and favors a joint distribution solution over a product of marginal distributions one. We apply CoGAN to several joint distribution learning tasks, including learning a joint distribution of color and depth images, and learning a joint distribution of face images with different attributes. For each task it successfully learns the joint distribution without any tuple of corresponding images. We also demonstrate its applications to domain adaptation and image transformation.

## 1 Introduction

The paper concerns the problem of learning a joint distribution of multi-domain images from data. A joint distribution of multi-domain images is a probability density function that gives a density value to each joint occurrence of images in different domains such as images of the same scene in different modalities (color and depth images) or images of the same face with different attributes (smiling and non-smiling). Once a joint distribution of multi-domain images is learned, it can be used to generate novel tuples of images. In addition to movie and game production, joint image distribution learning finds applications in image transformation and domain adaptation. When training data are given as tuples of corresponding images in different domains, several existing approaches [1, 2, 3, 4] can be applied. However, building a dataset with tuples of corresponding images is often a challenging task. This correspondence dependency greatly limits the applicability of the existing approaches.

To overcome the limitation, we propose the coupled generative adversarial networks (CoGAN) framework. It can learn a joint distribution of multi-domain images without existence of corresponding images in different domains in the training set. Only a set of images drawn separately from the marginal distributions of the individual domains is required. CoGAN is based on the generative adversarial networks (GAN) framework [5], which has been established as a viable solution for image distribution learning tasks. CoGAN extends GAN for joint image distribution learning tasks.

CoGAN consists of a tuple of GANs, each for one image domain. When trained naively, the CoGAN learns a product of marginal distributions rather than a joint distribution. We show that by enforcing a weight-sharing constraint the CoGAN can learn a joint distribution without existence of corresponding images in different domains. The CoGAN framework is inspired by the idea that deep neural networks learn a hierarchical feature representation. By enforcing the layers that decode high-level semantics in the GANs to share the weights, it forces the GANs to decode the high-level semantics in the same way. The layers that decode low-level details then map the shared representation to images in individual domains for confusing the respective discriminative models. CoGAN is for multi-image domains but, for ease of presentation, we focused on the case of two image domains in the paper. However, the discussions and analyses can be easily generalized to multiple image domains.

We apply CoGAN to several joint image distribution learning tasks. Through convincing visualization results and quantitative evaluations, we verify its effectiveness. We also show its applications to unsupervised domain adaptation and image transformation.

## 2   Generative Adversarial Networks

A GAN consists of a generative model and a discriminative model. The objective of the generative model is to synthesize images resembling real images, while the objective of the discriminative model is to distinguish real images from synthesized ones. Both the generative and discriminative models are realized as multilayer perceptrons.

Let $\mathbf{x}$ be a natural image drawn from a distribution, $p_X$, and $\mathbf{z}$ be a random vector in $\mathbb{R}^d$. Note that we only consider that $\mathbf{z}$ is from a uniform distribution with a support of $[-1\ 1]^d$, but different distributions such as a multivariate normal distribution can be applied as well. Let $g$ and $f$ be the generative and discriminative models, respectively. The generative model takes $\mathbf{z}$ as input and outputs an image, $g(\mathbf{z})$, that has the same support as $\mathbf{x}$. Denote the distribution of $g(\mathbf{z})$ as $p_G$. The discriminative model estimates the probability that an input image is drawn from $p_X$. Ideally, $f(\mathbf{x}) = 1$ if $\mathbf{x} \sim p_X$ and $f(\mathbf{x}) = 0$ if $\mathbf{x} \sim p_G$. The GAN framework corresponds to a minimax two-player game, and the generative and discriminative models can be trained jointly via solving

$$\max_g \min_f V(f, g) \equiv E_{\mathbf{x} \sim p_\mathbf{X}}[-\log f(\mathbf{x})] + E_{\mathbf{z} \sim p_\mathbf{z}}[-\log(1 - f(g(\mathbf{z})))]. \qquad (1)$$

In practice (1) is solved by alternating the following two gradient update steps:

$$\text{Step 1: } \boldsymbol{\theta}_f^{t+1} = \boldsymbol{\theta}_f^t - \lambda^t \nabla_{\boldsymbol{\theta}_f} V(f^t, g^t), \qquad \text{Step 2: } \boldsymbol{\theta}_g^{t+1} = \boldsymbol{\theta}_g^t + \lambda^t \nabla_{\boldsymbol{\theta}_g} V(f^{t+1}, g^t)$$

where $\boldsymbol{\theta}_f$ and $\boldsymbol{\theta}_g$ are the parameters of $f$ and $g$, $\lambda$ is the learning rate, and $t$ is the iteration number.

Goodfellow et al. [5] show that, given enough capacity to $f$ and $g$ and sufficient training iterations, the distribution, $p_G$, converges to $p_X$. In other words, from a random vector, $\mathbf{z}$, the network $g$ can synthesize an image, $g(\mathbf{z})$, that resembles one that is drawn from the true distribution, $p_X$.

## 3   Coupled Generative Adversarial Networks

CoGAN as illustrated in Figure 1 is designed for learning a joint distribution of images in two different domains. It consists of a pair of GANs—GAN$_1$ and GAN$_2$; each is responsible for synthesizing images in one domain. During training, we force them to share a subset of parameters. This results in that the GANs learn to synthesize pairs of corresponding images without correspondence supervision.

**Generative Models:**   Let $\mathbf{x}_1$ and $\mathbf{x}_2$ be images drawn from the marginal distribution of the 1st domain, $\mathbf{x}_1 \sim p_{X_1}$ and the marginal distribution of the 2nd domain, $\mathbf{x}_2 \sim p_{X_2}$, respectively. Let $g_1$ and $g_2$ be the generative models of GAN$_1$ and GAN$_2$, which map a random vector input $\mathbf{z}$ to images that have the same support as $\mathbf{x}_1$ and $\mathbf{x}_2$, respectively. Denote the distributions of $g_1(\mathbf{z})$ and $g_1(\mathbf{z})$ by $p_{G_1}$ and $p_{G_2}$. Both $g_1$ and $g_2$ are realized as multilayer perceptrons:

$$g_1(\mathbf{z}) = g_1^{(m_1)}\big(g_1^{(m_1-1)}\big(\dots g_1^{(2)}\big(g_1^{(1)}(\mathbf{z})\big)\big)\big), \quad g_2(\mathbf{z}) = g_2^{(m_2)}\big(g_2^{(m_2-1)}\big(\dots g_2^{(2)}\big(g_2^{(1)}(\mathbf{z})\big)\big)\big)$$

where $g_1^{(i)}$ and $g_2^{(i)}$ are the $i$th layers of $g_1$ and $g_2$ and $m_1$ and $m_2$ are the numbers of layers in $g_1$ and $g_2$. Note that $m_1$ need not equal $m_2$. Also note that the support of $\mathbf{x}_1$ need not equal to that of $\mathbf{x}_2$.

Through layers of perceptron operations, the generative models gradually decode information from more abstract concepts to more material details. The first layers decode high-level semantics and the last layers decode low-level details. Note that this information flow direction is opposite to that in a discriminative deep neural network [6] where the first layers extract low-level features while the last layers extract high-level features.

Based on the idea that a pair of corresponding images in two domains share the same high-level concepts, we force the first layers of $g_1$ and $g_2$ to have identical structure and share the weights. That is $\boldsymbol{\theta}_{g_1^{(i)}} = \boldsymbol{\theta}_{g_2^{(i)}}$, for $i = 1, 2, ..., k$ where $k$ is the number of shared layers, and $\boldsymbol{\theta}_{g_1^{(i)}}$ and $\boldsymbol{\theta}_{g_2^{(i)}}$ are the parameters of $g_1^{(i)}$ and $g_2^{(i)}$, respectively. This constraint forces the high-level semantics to be decoded in the same way in $g_1$ and $g_2$. No constraints are enforced to the last layers. They can materialize the shared high-level representation differently for fooling the respective discriminators.

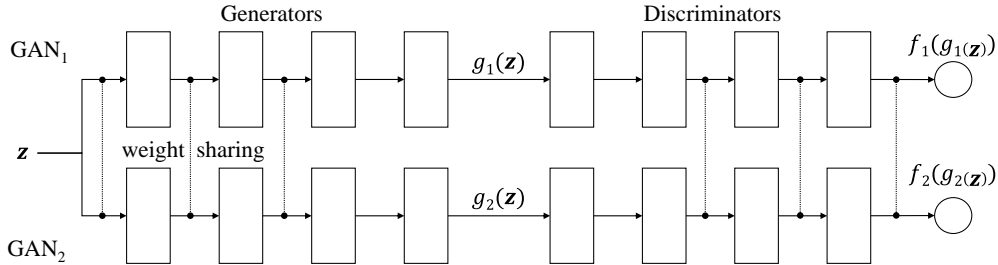

Figure 1: CoGAN consists of a pair of GANs: GAN$_1$ and GAN$_2$. Each has a generative model for synthesizing realistic images in one domain and a discriminative model for classifying whether an image is real or synthesized. We tie the weights of the first few layers (responsible for decoding high-level semantics) of the generative models, $g_1$ and $g_2$. We also tie the weights of the last few layers (responsible for encoding high-level semantics) of the discriminative models, $f_1$ and $f_2$. This weight-sharing constraint allows CoGAN to learn a joint distribution of images without correspondence supervision. A trained CoGAN can be used to synthesize pairs of corresponding images—pairs of images sharing the same high-level abstraction but having different low-level realizations.

**Discriminative Models:** Let $f_1$ and $f_2$ be the discriminative models of GAN$_1$ and GAN$_2$ given by

$$f_1(\mathbf{x}_1) = f_1^{(n_1)}\big(f_1^{(n_1-1)}\big(\ldots f_1^{(2)}\big(f_1^{(1)}(\mathbf{x}_1)\big)\big)\big), \ f_2(\mathbf{x}_2) = f_2^{(n_2)}\big(f_2^{(n_2-1)}\big(\ldots f_2^{(2)}\big(f_2^{(1)}(\mathbf{x}_2)\big)\big)\big)$$

where $f_1^{(i)}$ and $f_2^{(i)}$ are the $i$th layers of $f_1$ and $f_2$ and $n_1$ and $n_2$ are the numbers of layers. The discriminative models map an input image to a probability score, estimating the likelihood that the input is drawn from a true data distribution. The first layers of the discriminative models extract low-level features, while the last layers extract high-level features. Because the input images are realizations of the same high-level semantics in two different domains, we force $f_1$ and $f_2$ to have the same last layers, which is achieved by sharing the weights of the last layers via $\boldsymbol{\theta}_{f_1^{(n_1-i)}} = \boldsymbol{\theta}_{f_2^{(n_2-i)}}$, for $i = 0, 1, ..., l-1$ where $l$ is the number of weight-sharing layers in the discriminative models, and $\boldsymbol{\theta}_{f_1^{(i)}}$ and $\boldsymbol{\theta}_{f_2^{(i)}}$ are the network parameters of $f_1^{(i)}$ and $f_2^{(i)}$, respectively. The weight-sharing constraint in the discriminators helps reduce the total number of parameters in the network, but it is not essential for learning a joint distribution.

**Learning:** The CoGAN framework corresponds to a constrained minimax game given by

$$\max_{g_1,g_2} \min_{f_1,f_2} V(f_1, f_2, g_1, g_2), \ \text{subject to} \quad \boldsymbol{\theta}_{g_1^{(i)}} = \boldsymbol{\theta}_{g_2^{(i)}}, \qquad \text{for } i = 1, 2, ..., k \tag{2}$$

$$\boldsymbol{\theta}_{f_1^{(n_1-j)}} = \boldsymbol{\theta}_{f_2^{(n_2-j)}}, \ \text{for } j = 0, 1, ..., l-1$$

where the value function $V$ is given by

$$V(f_1, f_2, g_1, g_2) = E_{\mathbf{x}_1 \sim p_{\mathbf{X}_1}}[-\log f_1(\mathbf{x}_1)] + E_{\mathbf{z} \sim p_{\mathbf{Z}}}[-\log(1 - f_1(g_1(\mathbf{z})))]$$
$$+ E_{\mathbf{x}_2 \sim p_{\mathbf{X}_2}}[-\log f_2(\mathbf{x}_2)] + E_{\mathbf{z} \sim p_{\mathbf{Z}}}[-\log(1 - f_2(g_2(\mathbf{z})))]. \tag{3}$$

In the game, there are two teams and each team has two players. The generative models form a team and work together for synthesizing a pair of images in two different domains for confusing the discriminative models. The discriminative models try to differentiate images drawn from the training data distribution in the respective domains from those drawn from the respective generative models. The collaboration between the players in the same team is established from the weight-sharing constraint. Similar to GAN, CoGAN can be trained by back propagation with the alternating gradient update steps. The details of the learning algorithm are given in the supplementary materials.

**Remarks:** CoGAN learning requires training samples drawn from the marginal distributions, $p_{X_1}$ and $p_{X_2}$. It does not rely on samples drawn from the joint distribution, $p_{X_1, X_2}$, where corresponding supervision would be available. Our main contribution is in showing that with just samples drawn separately from the marginal distributions, CoGAN can learn a joint distribution of images in the two domains. Both weight-sharing constraint and adversarial training are essential for enabling this capability. Unlike autoencoder learning [3], which encourages a generated pair of images to be *identical* to the target pair of corresponding images in the two domains for minimizing the reconstruction loss[1], the adversarial training only encourages the generated pair of images to be

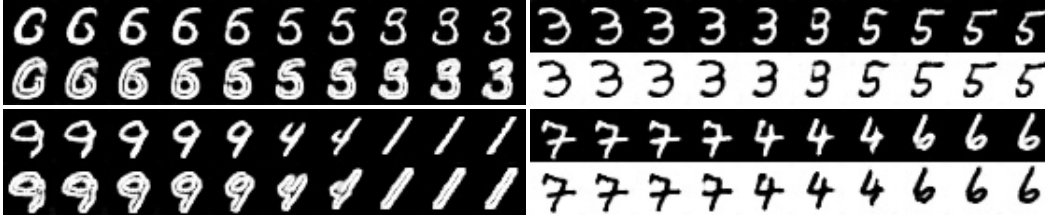

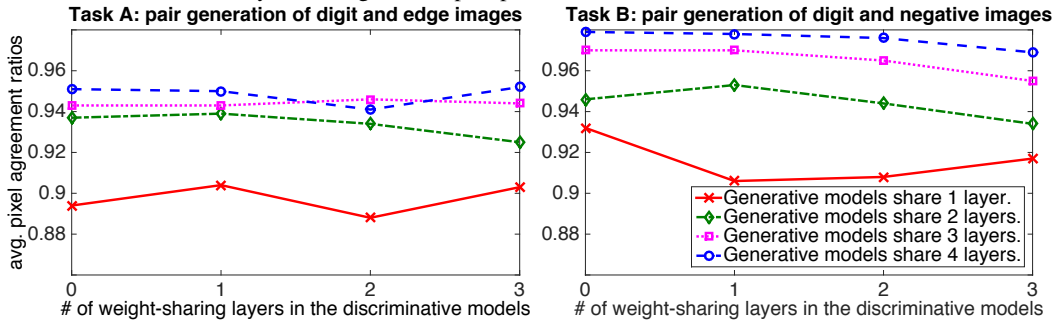

Figure 2: Left (Task $\mathbb{A}$): generation of digit and corresponding edge images. Right (Task $\mathbb{B}$): generation of digit and corresponding negative images. Each of the top and bottom pairs was generated using the same input noise. We visualized the results by traversing in the input space.

**Task A: pair generation of digit and edge images**     **Task B: pair generation of digit and negative images**

Figure 3: The figures plot the average pixel agreement ratios of the CoGANs with different weight-sharing configurations for Task $\mathbb{A}$ and $\mathbb{B}$. The larger the pixel agreement ratio the better the pair generation performance. We found that the performance was positively correlated with the number of weight-sharing layers in the generative models but was uncorrelated to the number of weight-sharing layers in the discriminative models. CoGAN learned the joint distribution without weight-sharing layers in the discriminative models.

*individually resembling to* the images in the respective domains. With this more relaxed adversarial training setting, the weight-sharing constraint can then kick in for capturing correspondences between domains. With the weight-sharing constraint, the generative models must utilize the capacity more efficiently for fooling the discriminative models, and the most efficient way of utilizing the capacity for generating a pair of realistic images in two domains is to generate a pair of *corresponding* images since the neurons responsible for decoding high-level semantics can be shared.

CoGAN learning is based on existence of shared high-level representations in the domains. If such a representation does not exist for the set of domains of interest, it would fail.

## 4 Experiments

In the experiments, we emphasized there were no corresponding images in the different domains in the training sets. CoGAN learned the joint distributions without correspondence supervision. We were unaware of existing approaches with the same capability and hence did not compare CoGAN with prior works. Instead, we compared it to a conditional GAN to demonstrate its advantage. Recognizing that popular performance metrics for evaluating generative models all subject to issues [7], we adopted a pair image generation performance metric for comparison. Many details including the network architectures and additional experiment results are given in the supplementary materials. An implementation of CoGAN is available in `https://github.com/mingyuliutw/cogan`.

**Digits:** We used the MNIST training set to train CoGANs for the following two tasks. Task $\mathbb{A}$ is about learning a joint distribution of a digit and its edge image. Task $\mathbb{B}$ is about learning a joint distribution of a digit and its negative image. In Task $\mathbb{A}$, the 1st domain consisted of the original handwritten digit images, while the 2nd domain consisted of their edge images. We used an edge detector to compute training edge images for the 2nd domain. In the supplementary materials, we also showed an experiment for learning a joint distribution of a digit and its 90-degree in-plane rotation.

We used deep convolutional networks to realized the CoGAN. The two generative models had an identical structure; both had 5 layers and were fully convolutional. The stride lengths of the convolutional layers were fractional. The models also employed the batch normalization processing [8] and the parameterized rectified linear unit processing [9]. We shared the parameters for all the layers except for the last convolutional layers. For the discriminative models, we used a variant of LeNet [10].

The inputs to the discriminative models were batches containing output images from the generative models and images from the two training subsets (each pixel value is linearly scaled to $[0\ 1]$).

We divided the training set into two equal-size *non-overlapping* subsets. One was used to train $GAN_1$ and the other was used to train $GAN_2$. We used the ADAM algorithm [11] for training and set the learning rate to 0.0002, the 1st momentum parameter to 0.5, and the 2nd momentum parameter to 0.999 as suggested in [12]. The mini-batch size was 128. We trained the CoGAN for 25000 iterations. These hyperparameters were fixed for all the visualization experiments.

The CoGAN learning results are shown in Figure 2. We found that although the CoGAN was trained without corresponding images, it learned to render corresponding ones for both Task $\mathbb{A}$ and $\mathbb{B}$. This was due to the weight-sharing constraint imposed to the layers that were responsible for decoding high-level semantics. Exploiting the correspondence between the two domains allowed $GAN_1$ and $GAN_2$ to utilize more capacity in the networks to better fit the training data. Without the weight-sharing constraint, the two GANs just generated two unrelated images in the two domains.

**Weight Sharing:** We varied the numbers of weight-sharing layers in the generative and discriminative models to create different CoGANs for analyzing the weight-sharing effect for both tasks. Due to lack of proper validation methods, we did a grid search on the training iteration hyperparameter and reported the best performance achieved by each network. For quantifying the performance, we transformed the image generated by $GAN_1$ to the 2nd domain using the same method employed for generating the training images in the 2nd domain. We then compared the transformed image with the image generated by $GAN_2$. A perfect joint distribution learning should render two identical images. Hence, we used the ratios of agreed pixels between 10K pairs of images generated by each network (10K randomly sampled $\mathbf{z}$) as the performance metric. We trained each network 5 times with different initialization weights and reported the average pixel agreement ratios over the 5 trials for each network. The results are shown in Figure 3. We observed that the performance was positively correlated with the number of weight-sharing layers in the generative models. With more sharing layers in the generative models, the rendered pairs of images resembled true pairs drawn from the joint distribution more. We also noted that the performance was uncorrelated to the number of weight-sharing layers in the discriminative models. However, we still preferred discriminator weight-sharing because this reduces the total number of network parameters.

**Comparison with Conditional GANs:** We compared the CoGAN with the conditional GANs [13]. We designed a conditional GAN with the generative and discriminative models identical to those in the CoGAN. The only difference was the conditional GAN took an additional binary variable as input, which controlled the domain of the output image. When the binary variable was 0, it generated an image resembling images in the 1st domain; otherwise, it generated an image resembling images in the 2nd domain. Similarly, no pairs of corresponding images were given during the conditional GAN training. We applied the conditional GAN to both Task $\mathbb{A}$ and $\mathbb{B}$ and hoped to empirically answer whether a conditional model can be used to learn to render corresponding images with correspondence supervision. The pixel agreement ratio was used as the performance metric. The experiment results showed that for Task $\mathbb{A}$, CoGAN achieved an average ratio of **0.952**, outperforming 0.909 achieved by the conditional GAN. For Task $\mathbb{B}$, CoGAN achieved a score of **0.967**, which was much better than 0.778 achieved by the conditional GAN. The conditional GAN just generated two different digits with the same random noise input but different binary variable values. These results showed that the conditional model failed to learn a joint distribution from samples drawn from the marginal distributions. We note that for the case that the supports of the two domains are different such as the color and depth image domains, the conditional model cannot even be applied.

**Faces:** We applied CoGAN to learn a joint distribution of face images with different. We trained several CoGANs, each for generating a face with an attribute and a corresponding face without the attribute. We used the CelebFaces Attributes dataset [14] for the experiments. The dataset covered large pose variations and background clutters. Each face image had several attributes, including blond hair, smiling, and eyeglasses. The face images with an attribute constituted the 1st domain; and those without the attribute constituted the 2nd domain. No corresponding face images between the two domains was given. We resized the images to a resolution of $132 \times 132$ and randomly sampled $128 \times 128$ regions for training. The generative and discriminative models were both 7 layer deep convolutional neural networks.

The experiment results are shown in Figure 4. We randomly sampled two points in the 100-dimensional input noise space and visualized the rendered face images as traveling from one pint to

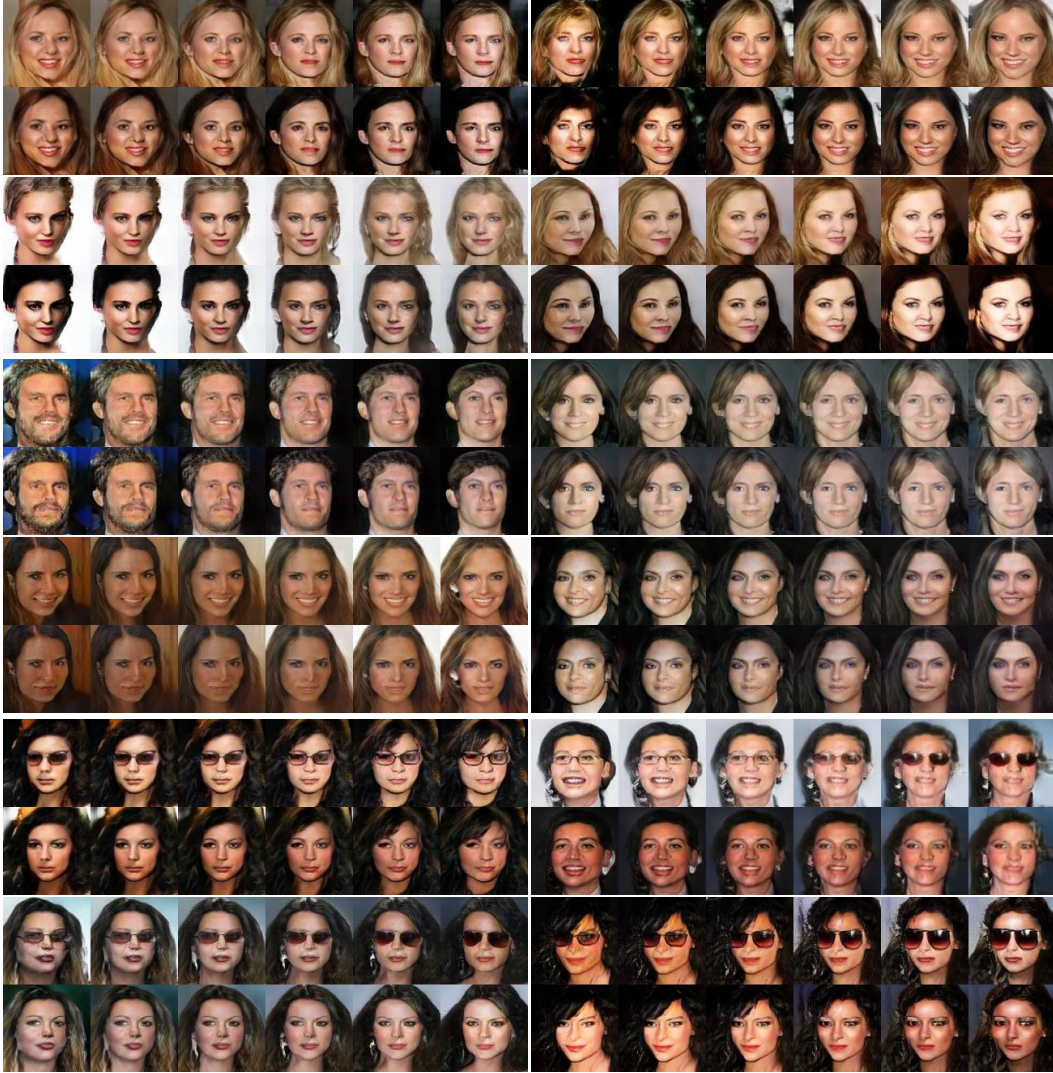

Figure 4: Generation of face images with different attributes using CoGAN. From top to bottom, the figure shows pair face generation results for the blond-hair, smiling, and eyeglasses attributes. For each pair, the 1st row contains faces with the attribute, while the 2nd row contains corresponding faces without the attribute.

the other. We found CoGAN generated pairs of corresponding faces, resembling those from the same person with and without an attribute. As traveling in the space, the faces gradually change from one person to another. Such deformations were consistent for both domains. Note that it is difficult to create a dataset with corresponding images for some attribute such as blond hair since the subjects have to color their hair. It is more ideal to have an approach that does not require corresponding images like CoGAN. We also noted that the number of faces with an attribute was often several times smaller than that without the attribute in the dataset. However, CoGAN learning was not hindered by the mismatches.

**Color and Depth Images:** We used the RGBD dataset [15] and the NYU dataset [16] for learning joint distribution of color and depth images. The RGBD dataset contains registered color and depth images of 300 objects captured by the Kinect sensor from different view points. We partitioned the dataset into two equal-size *non-overlapping* subsets. The color images in the 1st subset were used for training $GAN_1$, while the depth images in the 2nd subset were used for training $GAN_2$. There were no corresponding depth and color images in the two subsets. The images in the RGBD dataset have different resolutions. We resized them to a fixed resolution of $64 \times 64$. The NYU dataset contains color and depth images captured from indoor scenes using the Kinect sensor. We used the 1449 processed depth images for the depth domain. The training images for the color domain were from

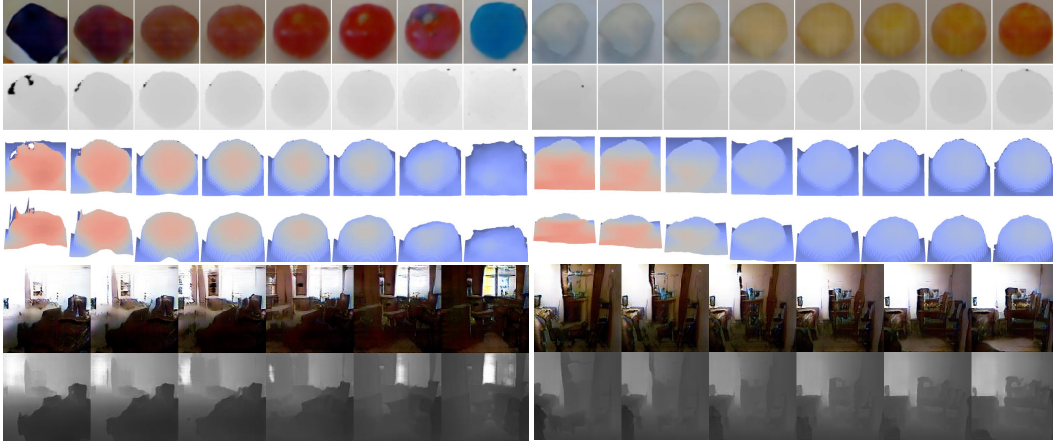

Figure 5: Generation of color and depth images using CoGAN. The top figure shows the results for the RGBD dataset: the 1st row contains the color images, the 2nd row contains the depth images, and the 3rd and 4th rows visualized the depth profile under different view points. The bottom figure shows the results for the NYU dataset.

all the color images in the raw dataset except for those registered with the processed depth images. We resized both the depth and color images to a resolution of $176 \times 132$ and randomly cropped $128 \times 128$ patches for training.

Figure 5 showed the generation results. We found the rendered color and depth images resembled corresponding RGB and depth image pairs despite of no registered images existed in the two domains in the training set. The CoGAN recovered the appearance–depth correspondence unsupervisedly.

## 5   Applications

In addition to rendering novel pairs of corresponding images for movie and game production, the CoGAN finds applications in the unsupervised domain adaptation and image transformation tasks.

**Unsupervised Domain Adaptation (UDA):** UDA concerns adapting a classifier trained in one domain to classify samples in a new domain where there is no labeled example in the new domain for re-training the classifier. Early works have explored ideas from subspace learning [17, 18] to deep discriminative network learning [19, 20, 21]. We show that CoGAN can be applied to the UDA problem. We studied the problem of adapting a digit classifier from the MNIST dataset to the USPS dataset. Due to domain shift, a classifier trained using one dataset achieves poor performance in the other. We followed the experiment protocol in [17, 20], which randomly samples 2000 images from the MNIST dataset, denoted as $D_1$, and 1800 images from the USPS dataset, denoted as $D_2$, to define an UDA problem. The USPS digits have a different resolution. We resized them to have the same resolution as the MNIST digits. We employed the CoGAN used for the digit generation task. For classifying digits, we attached a softmax layer to the last hidden layer of the discriminative models. We trained the CoGAN by jointly solving the digit classification problem in the MNIST domain which used the images and labels in $D_1$ and the CoGAN learning problem which used the images in both $D_1$ and $D_2$. This produced two classifiers: $c_1(\mathbf{x}_1) \equiv c(f_1^{(3)}(f_1^{(2)}(f_1^{(1)}(\mathbf{x}_1))))$ for MNIST and $c_2(\mathbf{x}_2) \equiv c(f_2^{(3)}(f_2^{(2)}(f_2^{(1)}(\mathbf{x}_2))))$ for USPS. No label information in $D_2$ was used. Note that $f_1^{(2)} \equiv f_2^{(2)}$ and $f_1^{(3)} \equiv f_2^{(3)}$ due to weight sharing and $c$ denotes the softmax layer. We then applied $c_2$ to classify digits in the USPS dataset. The classifier adaptation from USPS to MNIST can be achieved in the same way. The learning hyperparameters were determined via a validation set. We reported the average accuracy over 5 trails with different randomly selected $D_1$ and $D_2$.

Table 1 reports the performance of the proposed CoGAN approach with comparison to the state-of-the-art methods for the UDA task. The results for the other methods were duplicated from [20]. We observed that CoGAN significantly outperformed the state-of-the-art methods. It improved the accuracy from 0.64 to 0.90, which translates to a **72%** error reduction rate.

**Cross-Domain Image Transformation:** Let $\mathbf{x}_1$ be an image in the 1st domain. Cross-domain image transformation is about finding the corresponding image in the 2nd domain, $\mathbf{x}_2$, such that the joint

| Method | [17] | [18] | [19] | [20] | CoGAN |
|---|---|---|---|---|---|
| From MNIST to USPS | 0.408 | 0.467 | 0.478 | 0.607 | **0.912** ±0.008 |
| From USPS to MNIST | 0.274 | 0.355 | 0.631 | 0.673 | **0.891** ±0.008 |
| Average | 0.341 | 0.411 | 0.554 | 0.640 | **0.902** |

Table 1: Unsupervised domain adaptation performance comparison. The table reported classification accuracies achieved by competing algorithms.

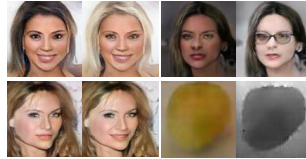

Figure 6: Cross-domain image transformation. For each pair, left is the input; right is the transformed image.

probability density, $p(\mathbf{x}_1, \mathbf{x}_2)$, is maximized. Let $\mathcal{L}$ be a loss function measuring difference between two images. Given $g_1$ and $g_2$, the transformation can be achieved by first finding the random vector that generates the query image in the 1st domain $\mathbf{z}^* = arg\min_{\mathbf{z}} \mathcal{L}(g_1(\mathbf{z}), \mathbf{x}_1)$. After finding $\mathbf{z}^*$, one can apply $g_2$ to obtain the transformed image, $\mathbf{x}_2 = g_2(\mathbf{z}^*)$. In Figure 6, we show several CoGAN cross-domain transformation results, computed by using the Euclidean loss function and the L-BFGS optimization algorithm. We found the transformation was successful when the input image was covered by $g_1$ (The input image can be generated by $g_1$.) but generated blurry images when it is not the case. To improve the coverage, we hypothesize that more training images and a better objective function are required, which are left as future work.

## 6   Related Work

Neural generative models has recently received an increasing amount of attention. Several approaches, including generative adversarial networks[5], variational autoencoders (VAE)[22], attention models[23], moment matching[24], stochastic back-propagation[25], and diffusion processes[26], have shown that a deep network can learn an image distribution from samples. The learned networks can be used to generate novel images. Our work was built on [5]. However, we studied a different problem, the problem of learning a *joint* distribution of multi-domain images. We were interested in whether a joint distribution of images in different domains can be learned from samples drawn separately from its marginal distributions of the individual domains. We showed its achievable via the proposed CoGAN framework. Note that our work is different to the Attribute2Image work[27], which is based on a conditional VAE model [28]. The conditional model can be used to generate images of different styles, but they are unsuitable for generating images in two different domains such as color and depth image domains.

Following [5], several works improved the image generation quality of GAN, including a Laplacian pyramid implementation[29], a deeper architecture[12], and conditional models[13]. Our work extended GAN to dealing with joint distributions of images.

Our work is related to the prior works in multi-modal learning, including joint embedding space learning [30] and multi-modal Boltzmann machines [1, 3]. These approaches can be used for generating corresponding samples in different domains only when correspondence annotations are given during training. The same limitation is also applied to dictionary learning-based approaches [2, 4]. Our work is also related to the prior works in cross-domain image generation [31, 32, 33], which studied transforming an image in one style to the corresponding images in another style. However, we focus on learning the joint distribution in an unsupervised fashion, while [31, 32, 33] focus on learning a transformation function directly in a supervised fashion.

## 7   Conclusion

We presented the CoGAN framework for learning a joint distribution of multi-domain images. We showed that via enforcing a simple weight-sharing constraint to the layers that are responsible for decoding abstract semantics, the CoGAN learned the joint distribution of images by just using samples drawn separately from the marginal distributions. In addition to convincing image generation results on faces and RGBD images, we also showed promising results of the CoGAN framework for the image transformation and unsupervised domain adaptation tasks.

## Footnotes

[1]This is why [3] requires samples from the joint distribution for learning the joint distribution.

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
