[Supplementary Material]

# A    Additional Experiment Results

## A.1    Rotation

We applied CoGAN to a task of learning a joint distribution of images with different in-plane rotation angles. We note that this task is very different to the other tasks discussed in the paper. In the other tasks, the image contents in the same spatial region in the corresponding images are in direct correspondence. In this task, the content in one spatial region in one image domain is related to the content in a different spatial region in the other image domain. Through this experiment, we planed to verify whether CoGAN can learn a joint distribution of images related by a global transformation.

For this task, we partitioned the MNIST training set into two disjoint subsets. The first set consisted of the original digit images, which constitute the first domain. We applied a 90 degree rotation to all the digits in the second set to construct the second domain. There were no corresponding images in the two domains. The CoGAN architecture used for this task is shown in Table 1. Different to the other tasks, the generative models in the CoGAN were based on fully connected layers, and the discriminative models only share the last layer. This design was due to lack of spatial correspondence between the two domains. We used the same hyperparameters to train the CoGAN. The results are shown in Figure 1. We found that the CoGAN was able to capture the in-plane rotation. For the same noise input, the digit generated by $GAN_2$ is a 90 degree rotated version of the digit generated by $GAN_1$.

Table 1: CoGAN for generating digits with different in-plane rotation angles

| Generative models | | | |
|---|---|---|---|
| Layer | Domain 1 | Domain 2 | Shared? |
| 1 | FC-(N1024), BN, PReLU | FC-(N1024), BN, PReLU | Yes |
| 2 | FC-(N1024), BN, PReLU | FC-(N1024), BN, PReLU | Yes |
| 3 | FC-(N1024), BN, PReLU | FC-(N1024), BN, PReLU | Yes |
| 4 | FC-(N1024), BN, PReLU | FC-(N1024), BN, PReLU | Yes |
| 5 | FC-(N784), Sigmoid | FC-(N784), Sigmoid | No |
| Discriminative models | | | |
| Layer | Domain 1 | Domain 2 | Shared? |
| 1 | CONV-(N20,K5x5,S1), POOL-(MAX,2) | CONV-(N20,K5x5,S1), POOL-(MAX,2) | No |
| 2 | CONV-(N50,K5x5,S1), POOL-(MAX,2) | CONV-(N50,K5x5,S1), POOL-(MAX,2) | No |
| 3 | FC-(N500), PReLU | FC-(N500), PReLU | No |
| 4 | FC-(N1), Sigmoid | FC-(N1), Sigmoid | Yes |

Figure 1: Generation of digit and 90-degree rotated digit images. We visualized the CoGAN results by rendering pairs of images, using the vectors that corresponded to paths connecting two pints in the input noise space. For each of the sub-figures, the top row was from $GAN_1$ and the bottom row was from $GAN_2$. Each of the top and bottom pairs was rendered using the same input noise vector. We observed that CoGAN learned to synthesized corresponding digits with different rotation angles.

## A.2    Weight Sharing

We analyzed the effect of weight sharing in the CoGAN framework. We conducted an experiment where we varied the numbers of weight-sharing layers in the generative and discriminative models

to create different CoGAN architectures and trained them with the same hyperparameters. Due to lack of proper validation methods, we did a grid search on the training iteration and reported the best performance achieved by each network configuration for both Task $\mathbb{A}$ and $\mathbb{B}$[1]. For each network architecture, we run 5 trails with different random network initialization weights. We then rendered 10000 pairs of images for each learned network. A pair of images consisted of an image in the first domain (generated by $GAN_1$) and an image in the second domain (generated by $GAN_2$), which were rendered using the same $\mathbf{z}$.

For quantifying the performance of each weight-sharing scheme, we transformed the images generated by $GAN_1$ to the second domain by using the same method employed for generating the training images in the second domain. We then compared the transformed images with the images generated by $GAN_2$. The performance was measured by the average of the ratios of agreed pixels between the transformed image and the corresponding image in the other domain. Specifically, we rounded the transformed digit image to a binary image and we also rounded the rendered image in the second domain to a binary image. We then compared the pixel agreement ratio—the number of corresponding pixels that have the same value in the two images divided by the total image size. The performance of a trail was given by the pixel agreement ratio of the 10000 pairs of images. The performance of a network configuration was given by the average pixel agreement ratio over the 5 trails. We reported the performance results for Task $\mathbb{A}$ in Table 2 and the performance results for Task $\mathbb{B}$ in Table 3.

From the tables, we observed that the pair image generation performance was positively correlated with the number of weight-sharing layers in the generative models. With more shared layers in the generative models, the rendered pairs of images were resembling more to true pairs drawn from the joint distribution. We noted that the pair image generation performance was uncorrelated to the number of weight-sharing layers in the discriminative models. However, we still preferred applying discriminator weight sharing because this reduces the total number of parameters.

Table 2: The table shows the performance of pair generation of digits and corresponding edge images (Task $\mathbb{A}$) with different CoGAN weight-sharing configurations. The results were the average pixel agreement ratios over 10000 images over 5 trials.

| Avg. pixel agreement ratio | | Weight-sharing layers in the generative models | | | |
| --- | --- | --- | --- | --- | --- |
| | | 5 | 5,4 | 5,4,3 | 5,4,3,2 |
| Weight-sharing | | $0.894 \pm 0.020$ | $0.937 \pm 0.004$ | $0.943 \pm 0.003$ | $0.951 \pm 0.004$ |
| layers in the | 4 | $0.904 \pm 0.018$ | $0.939 \pm 0.002$ | $0.943 \pm 0.005$ | $0.950 \pm 0.003$ |
| discriminative | 4,3 | $0.888 \pm 0.036$ | $0.934 \pm 0.005$ | $0.946 \pm 0.003$ | $0.941 \pm 0.024$ |
| models | 4,3,2 | $0.903 \pm 0.009$ | $0.925 \pm 0.021$ | $0.944 \pm 0.006$ | $0.952 \pm 0.002$ |

Table 3: The table shows the performance of pair generation of digits and corresponding negative images (Task $\mathbb{B}$) with different CoGAN weight-sharing configurations. The results were the average pixel agreement ratios over 10000 images over 5 trials.

| Avg. pixel agreement ratio | | Weight-sharing layers in the generative models | | | |
| --- | --- | --- | --- | --- | --- |
| | | 5 | 5,4 | 5,4,3 | 5,4,3,2 |
| Weight-sharing | | $0.932 \pm 0.011$ | $0.946 \pm 0.013$ | $0.970 \pm 0.002$ | $0.979 \pm 0.001$ |
| layers in the | 4 | $0.906 \pm 0.066$ | $0.953 \pm 0.008$ | $0.970 \pm 0.003$ | $0.978 \pm 0.001$ |
| discriminative | 4,3 | $0.908 \pm 0.028$ | $0.944 \pm 0.012$ | $0.965 \pm 0.009$ | $0.976 \pm 0.001$ |
| models | 4,3,2 | $0.917 \pm 0.022$ | $0.934 \pm 0.011$ | $0.955 \pm 0.010$ | $0.969 \pm 0.008$ |

## A.3 Comparison with the Conditional Generative Adversarial Nets

We compared the CoGAN framework with the conditional generative adversarial networks (GAN) framework for joint image distribution learning. We designed a conditional GAN where the generative and discriminative models were identical to those used in the CoGAN in the digit experiments. The only difference was that the conditional GAN took an additional binary variable as input, which controlled the domain of the output image. The binary variable acted as a switch. When the value of the binary variable was zero, it generated images resembling images in the first domain. Otherwise, it generated images resembling those in the second domain. The output layer of the discriminative

Table 4: Network architecture of the conditional GAN

| Layer | Generative models |
|---|---|
| input | $\mathbf{z}$ and conditional variable $c \in \{0, 1\}$ |
| 1 | FCONV-(N1024,K4x4,S1), BN, PReLU |
| 2 | FCONV-(N512,K3x3,S2), BN, PReLU |
| 3 | FCONV-(N256,K3x3,S2), BN, PReLU |
| 4 | FCONV-(N128,K3x3,S2), BN, PReLU |
| 5 | FCONV-(N1,K6x6,S1), Sigmoid |

| Layer | Discriminative models |
|---|---|
| 1 | CONV-(N20,K5x5,S1), POOL-(MAX,2) |
| 2 | CONV-(N50,K5x5,S1), POOL-(MAX,2) |
| 3 | FC-(N500), PReLU |
| 4 | FC-(N3), Softmax |

Table 5: Performance Comparison. For each task, we reported the average pixel agreement ratio scores and standard deviations over 5 trails, each trained with a different random initialization of the network connection weights.

| Experiment | Task $\mathbb{A}$: Digit and Edge Images | Task $\mathbb{B}$: Digit and Negative Images |
|---|---|---|
| Conditional GAN | $0.909 \pm 0.003$ | $0.778 \pm 0.021$ |
| CoGAN | $\mathbf{0.952} \pm 0.002$ | $\mathbf{0.967} \pm 0.008$ |

Figure 2: Digit Generation with **Conditional** Generative Adversarial Nets. Left: generation of digit and corresponding edge images. Right: generation of digit and corresponding negative images. We visualized the conditional GAN results by rendering pairs of images, using the vectors that corresponded to paths connecting two pints in the input space. For each of the sub-figures, the top row was from the conditional GAN with the conditional variable set to 0, and the bottom row was from the conditional GAN with the conditional variable set to 1. That is each of the top and bottom pairs was rendered using the same input vector except for the conditional variable value. The conditional variable value was used to control the domain of the output images. From the figure, we observed that, although the conditional GAN learned to generate realistic digit images, it failed to learn the correspondence in the two domains. For the edge task, the conditional GAN rendered images of the same digits with a similar font. The edge style was not well-captured. For the negative image generation task, the conditional GAN simply failed to capture any correspondence. The rendered digits with the same input vector but different conditional variable values were not related.

model was a softmax layer with three neurons. If the first neuron was on, it meant the input to the discriminative model was a synthesized image from the generative model. If the second neuron was

on, it meant the input was a real image from the first domain. If the third neuron was on, it meant the input was a real image from the second domain. The goal of the generative model was to render images resembling those from the first domain when the binary variable was zero and to render images resembling those from the second domain when the binary variable was one. The details of the conditional GAN network architecture is shown in Table 4.

Similarly to CoGAN learning, no correspondence was given during the conditional GAN learning. We applied the conditional GAN to the two digit generation tasks and hoped to answer whether a conditional model can be used to render corresponding images in two different domains without pairs of corresponding images in the training set. We used the same training data and hyperparameters as those used in the CoGAN learning. We trained the CoGAN for 25000 iterations[2] and used the trained network to render 10000 pairs of images in the two domains. Specifically, each pair of images was rendered with the same $\mathbf{z}$ but with different conditional variable values. These images were used to compute the pair image generation performance of the conditional GAN measured by the average of the pixel agreement ratios. For each task, we trained the conditional GAN for 5 times, each with a different random initialization of the network weights. We reported the average scores and the standard deviations.

The performance results are reported in Table 5. It can be seen that the conditional GAN achieved 0.909 for Task $\mathbb{A}$ and 0.778 for Task $\mathbb{B}$, respectively. They were much lower than the scores of 0.952 and 0.967 achieved by the CoGAN. Figure 2 visualized the conditional GAN's pair generation results, which suggested that the conditional GAN had difficulties in learning to render corresponding images in two different domains without pairs of corresponding images in the training set.

# B CoGAN Learning Algorithm

We present the learning algorithm for the coupled generative adversarial networks in Algorithm 1. The algorithm is an extension of the learning algorithm for the generative adversarial networks (GAN) to the case of training two GANs with weight sharing constraints. The convergence property follows the results shown in [1].

---

**Algorithm 1** Mini-batch stochastic gradient descent for training coupled generative adversarial nets.

---

1: Initialize the network parameters $\boldsymbol{\theta}_{f_1^{(i)}}$'s $\boldsymbol{\theta}_{f_2^{(i)}}$'s $\boldsymbol{\theta}_{g_1^{(i)}}$'s and $\boldsymbol{\theta}_{g_2^{(i)}}$'s with the shared network connection weights set to the same values.

2: **for** $t = 0, 1, 2, ...,$ maximum number of iterations **do**

3:      Draw $N$ samples from $p_Z$, $\{\mathbf{z}^1, \mathbf{z}^2, ..., \mathbf{z}^N\}$

4:      Draw $N$ samples from $p_{X_1}$, $\{\mathbf{x}_1^1, \mathbf{x}_1^2, ..., \mathbf{x}_1^N\}$

5:      Draw $N$ samples from $p_{X_2}$, $\{\mathbf{x}_2^1, \mathbf{x}_2^2, ..., \mathbf{x}_2^N\}$

6:      Compute the gradients of the parameters of the discriminative model, $f_1^t$, $\Delta\boldsymbol{\theta}_{f_1^{(i)}}$;

$$\nabla_{\boldsymbol{\theta}_{f_1^{(i)}}} \frac{1}{N} \sum_{j=1}^{N} -\log f_1^t(\mathbf{x}_1^j) - \log\left(1 - f_1^t\big(g_1^t(\mathbf{z}^j)\big)\right)$$

7:      Compute the gradients of the parameters of the discriminative model, $f_2^t$, $\Delta\boldsymbol{\theta}_{f_2^{(i)}}$;

$$\nabla_{\boldsymbol{\theta}_{f_2^{(i)}}} \frac{1}{N} \sum_{j=1}^{N} -\log f_2^t(\mathbf{x}_2^j) - \log\left(1 - f_2^t\big(g_2^t(\mathbf{z}^j)\big)\right)$$

8:      Average the gradients of the shared parameters of the discriminative models.

9:      Compute $f_1^{t+1}$ and $f_2^{t+1}$ according to the gradients.

10:      Compute the gradients of the parameters of the generative model, $g_1^t$, $\Delta\boldsymbol{\theta}_{g_1^{(i)}}$;

$$\nabla_{\boldsymbol{\theta}_{g_1^{(i)}}} \frac{1}{N} \sum_{j=1}^{N} -\log\left(1 - f_1^{t+1}\big(g_1^t(\mathbf{z}^j)\big)\right)$$

11:      Compute the gradients of the network parameters of the generative model, $g_2$, $\Delta\boldsymbol{\theta}_{g_2^{(i)}}$;

$$\nabla_{\boldsymbol{\theta}_{g_2^{(i)}}} \frac{1}{N} \sum_{j=1}^{N} -\log\left(1 - f_2^{t+1}\big(g_2^t(\mathbf{z}^j)\big)\right)$$

12:      Average the gradients of the shared parameters of the generative models.

13:      Compute $g_1^{t+1}$ and $g_2^{t+1}$ according to the gradients.

14: **end for**

---

# C Training Datasets

In Figure 3, Figure 4, Figure 5, and Figure 6, we show several example images of the training images used for the pair image generation tasks in the experiment section. Table 6, Table 7, Table 8, and Table 9 contain the statistics of the training datasets for the experiments.

Figure 3: Training images for the digit experiments. Left (Task $\mathbb{A}$): The images in the first row are from the original MNIST digit domain, while those in the second row are from the edge image domain. Right (Task $\mathbb{B}$): The images in the first row are from the original MNIST digit domain, while those in the second row are from the negative image domain.

Figure 4: Training images from the Celeba dataset [2].

Figure 5: Training images from the RGBD dataset [3].

Figure 6: Training images from the NYU dataset [4].

Table 6: Numbers of training images in Domain 1 and Domain 2 in the MNIST experiments.

|  | Task $\mathbb{A}$ <br> Pair generation of digits and corresponding edge images | Task $\mathbb{B}$ <br> Pair generation of digits and corresponding negative images |
|---|---|---|
| # of images in Domain 1 | 30,000 | 30,000 |
| # of images in Domain 2 | 30,000 | 30,000 |

Table 7: Numbers of training images of different attributes in the pair face generation experiments.

| Attribute | Smiling | Blond hair | Glasses |
|---|---|---|---|
| # of images with the attribute | 97,669 | 29,983 | 13,193 |
| # of images without the attribute | 104,930 | 172,616 | 189,406 |

Table 8: Numbers of RGB and depth training images in the RGBD experiments.

| | |
|---|---|
| # of RGB images | 93,564 |
| # of depth images | 93,564 |

Table 9: Numbers of RGB and depth training images in the NYU experiments.

| | |
|---|---|
| # of RGB images | 514,192 |
| # of depth images | 1,449 |

# D   Networks

In CoGAN, the generative models are based on the fractional length convolutional (FCONV) layers, while the discriminative models are based on the standard convolutional (CONV) layers with the exceptions that the last two layers are based on the fully-connected (FC) layers. The batch normalization (BN) layers [5] are applied after each convolutional layer, which are followed by the parameterized rectified linear unit (PReLU) processing [6]. The sigmoid units and the hyperbolic tangent units are applied to the output layers of the generative models for generating images with desired pixel range values.

Table 10: CoGAN for digit generation

| Generative models | | | |
|---|---|---|---|
| Layer | Domain 1 | Domain 2 | Shared? |
| 1 | FCONV-(N1024,K4x4,S1), BN, PReLU | FCONV-(N1024,K4x4,S1), BN, PReLU | Yes |
| 2 | FCONV-(N512,K3x3,S2), BN, PReLU | FCONV-(N512,K3x3,S2), BN, PReLU | Yes |
| 3 | FCONV-(N256,K3x3,S2), BN, PReLU | FCONV-(N256,K3x3,S2), BN, PReLU | Yes |
| 4 | FCONV-(N128,K3x3,S2), BN, PReLU | FCONV-(N128,K3x3,S2), BN, PReLU | Yes |
| 5 | FCONV-(N1,K6x6,S1), Sigmoid | FCONV-(N1,K6x6,S1), Sigmoid | No |
| Discriminative models | | | |
| Layer | Domain 1 | Domain 2 | Shared? |
| 1 | CONV-(N20,K5x5,S1), POOL-(MAX,2) | CONV-(N20,K5x5,S1), POOL-(MAX,2) | No |
| 2 | CONV-(N50,K5x5,S1), POOL-(MAX,2) | CONV-(N50,K5x5,S1), POOL-(MAX,2) | Yes |
| 3 | FC-(N500), PReLU | FC-(N500), PReLU | Yes |
| 4 | FC-(N1), Sigmoid | FC-(N1), Sigmoid | Yes |

Table 11: CoGAN for face generation

| Generative models | | | |
|---|---|---|---|
| Layer | Domain 1 | Domain 2 | Shared? |
| 1 | FCONV-(N1024,K4x4,S1), BN, PReLU | FCONV-(N1024,K4x4,S1), BN, PReLU | Yes |
| 2 | FCONV-(N512,K4x4,S2), BN, PReLU | FCONV-(N512,K4x4,S2), BN, PReLU | Yes |
| 3 | FCONV-(N256,K4x4,S2), BN, PReLU | FCONV-(N256,K4x4,S2), BN, PReLU | Yes |
| 4 | FCONV-(N128,K4x4,S2), BN, PReLU | FCONV-(N128,K4x4,S2), BN, PReLU | Yes |
| 5 | FCONV-(N64,K4x4,S2), BN, PReLU | FCONV-(N64,K4x4,S2), BN, PReLU | Yes |
| 6 | FCONV-(N32,K4x4,S2), BN, PReLU | FCONV-(N32,K4x4,S2), BN, PReLU | No |
| 7 | FCONV-(N3,K3x3,S1), TanH | FCONV-(N3,K3x3,S1), TanH | No |
| Discriminative models | | | |
| Layer | Domain 1 | Domain 2 | Shared? |
| 1 | CONV-(N32,K5x5,S2), BN, PReLU | CONV-(N32,K5x5,S2), BN, PReLU | No |
| 2 | CONV-(N64,K5x5,S2), BN, PReLU | CONV-(N64,K5x5,S2), BN, PReLU | No |
| 3 | CONV-(N128,K5x5,S2), BN, PReLU | CONV-(N128,K5x5,S2), BN, PReLU | Yes |
| 4 | CONV-(N256,K3x3,S2), BN, PReLU | CONV-(N256,K3x3,S2), BN, PReLU | Yes |
| 5 | CONV-(N512,K3x3,S2), BN, PReLU | CONV-(N512,K3x3,S2), BN, PReLU | Yes |
| 6 | CONV-(N1024,K3x3,S2), BN, PReLU | CONV-(N1024,K3x3,S2), BN, PReLU | Yes |
| 7 | FC-(N2048), BN, PReLU | FC-(N2048), BN, PReLU | Yes |
| 8 | FC-(N1), Sigmoid | FC-(N1), Sigmoid | Yes |

Table 12: CoGAN for color and depth image generation for the RGBD object dataset

| Generative models | | | |
|---|---|---|---|
| Layer | Domain 1 | Domain 2 | Shared? |
| 1 | FCONV-(N1024,K4x4,S1), BN, PReLU | FCONV-(N1024,K4x4,S1), BN, PReLU | Yes |
| 2 | FCONV-(N512,K4x4,S2), BN, PReLU | FCONV-(N512,K4x4,S2), BN, PReLU | Yes |
| 3 | FCONV-(N256,K4x4,S2), BN, PReLU | FCONV-(N256,K4x4,S2), BN, PReLU | Yes |
| 4 | FCONV-(N128,K4x4,S2), BN, PReLU | FCONV-(N128,K4x4,S2), BN, PReLU | Yes |
| 5 | FCONV-(N64,K4x4,S2), BN, PReLU | FCONV-(N64,K4x4,S2), BN, PReLU | Yes |
| 6 | FCONV-(N32,K3x3,S1), BN, PReLU | FCONV-(N32,K3x3,S1), BN, PReLU | No |
| 7 | FCONV-(N3,K3x3,S1), TanH | FCONV-(N1,K3x3,S1), Sigmoid | No |
| Discriminative models | | | |
| Layer | Domain 1 | Domain 2 | Shared? |
| 1 | CONV-(N32,K5x5,S2), BN, PReLU | CONV-(N32,K5x5,S2), BN, PReLU | No |
| 2 | CONV-(N64,K5x5,S2), BN, PReLU | CONV-(N64,K5x5,S2), BN, PReLU | No |
| 3 | CONV-(N128,K5x5,S2), BN, PReLU | CONV-(N128,K5x5,S2), BN, PReLU | Yes |
| 4 | CONV-(N256,K3x3,S2), BN, PReLU | CONV-(N256,K3x3,S2), BN, PReLU | Yes |
| 5 | CONV-(N512,K3x3,S2), BN, PReLU | CONV-(N512,K3x3,S2), BN, PReLU | Yes |
| 6 | CONV-(N1024,K3x3,S2), BN, PReLU | CONV-(N1024,K3x3,S2), BN, PReLU | Yes |
| 7 | FC-(N2048), BN, PReLU | FC-(N2048), BN, PReLU | Yes |
| 8 | FC-(N1), Sigmoid | FC-(N1), Sigmoid | Yes |

Table 13: CoGAN for color and depth image generation for the NYU indoor scene dataset

| Generative models | | | |
|---|---|---|---|
| Layer | Domain 1 | Domain 2 | Shared? |
| 1 | FCONV-(N1024,K4x4,S1), BN, PReLU | FCONV-(N1024,K4x4,S1), BN, PReLU | Yes |
| 2 | FCONV-(N512,K4x4,S2), BN, PReLU | FCONV-(N512,K4x4,S2), BN, PReLU | Yes |
| 3 | FCONV-(N256,K4x4,S2), BN, PReLU | FCONV-(N256,K4x4,S2), BN, PReLU | Yes |
| 4 | FCONV-(N128,K4x4,S2), BN, PReLU | FCONV-(N128,K4x4,S2), BN, PReLU | Yes |
| 5 | FCONV-(N64,K4x4,S2), BN, PReLU | FCONV-(N64,K4x4,S2), BN, PReLU | Yes |
| 6 | FCONV-(N32,K4x4,S2), BN, PReLU | FCONV-(N32,K4x4,S2), BN, PReLU | No |
| 7 | FCONV-(N3,K3x3,S1), TanH | FCONV-(N1,K3x3,S1), Sigmoid | No |
| Discriminative models | | | |
| Layer | Domain 1 | Domain 2 | Shared? |
| 1 | CONV-(N32,K5x5,S2), BN, PReLU | CONV-(N32,K5x5,S2), BN, PReLU | No |
| 2 | CONV-(N64,K5x5,S2), BN, PReLU | CONV-(N64,K5x5,S2), BN, PReLU | No |
| 3 | CONV-(N128,K5x5,S2), BN, PReLU | CONV-(N128,K5x5,S2), BN, PReLU | Yes |
| 4 | CONV-(N256,K3x3,S2), BN, PReLU | CONV-(N256,K3x3,S2), BN, PReLU | Yes |
| 5 | CONV-(N512,K3x3,S2), BN, PReLU | CONV-(N512,K3x3,S2), BN, PReLU | Yes |
| 6 | CONV-(N1024,K3x3,S2), BN, PReLU | CONV-(N1024,K3x3,S2), BN, PReLU | Yes |
| 7 | FC-(N2048), BN, PReLU | FC-(N2048), BN, PReLU | Yes |
| 8 | FC-(N1), Sigmoid | FC-(N1), Sigmoid | Yes |

# E   Visualization

Figure 7: Left: generation of digit and corresponding edge images. Right: generation of digit and corresponding negative images. We visualized the CoGAN results by rendering pairs of images, using the vectors that corresponded to paths connecting two pints in the input noise space. For each of the sub-figures, the top row was from $GAN_1$ and the bottom row was from $GAN_2$. Each of the top and bottom pairs was rendered using the same input noise vector. We observed that for both tasks the CoGAN learned to synthesized corresponding images in the two domains. This was interesting because there were no corresponding images in the training datasets. The correspondences were figured out during training in an unsupervised fashion.

Figure 8: Generation of faces with blond hair and without blond hair.

Figure 9: Generation of faces with blond hair and without blond hair.

Figure 10: Generation of faces with blond hair and without blond hair.

Figure 11: Generation of smiling and non-smiling faces.

Figure 12: Generation of smiling and non-smiling faces.

Figure 13: Generation of smiling and non-smiling faces.

Figure 14: Generation of faces with eyeglasses and without eyeglasses.

Figure 15: Generation of faces with eyeglasses and without eyeglasses.

Figure 16: Generation of faces with eyeglasses and without eyeglasses.

Figure 17: Generation of RGB and depth images of objects. The 1st row contains the color images. The 2nd row contains the depth images. The 3rd and 4th rows visualized the point clouds under different view points.

Figure 18: Generation of RGB and depth images of objects. The 1st row contains the color images. The 2nd row contains the depth images. The 3rd and 4th rows visualized the point clouds under different view points.

Figure 19: Generation of RGB and depth images of indoor scenes.

Figure 20: Generation of RGB and depth images of indoor scenes.

## Footnotes

[1]We noted that the performances were not sensitive to the number of training iterations.

[2] We note the generation performance of the conditional GAN did not change much after 5000 iterations.