[Reviews · NeurIPS 2016]

Reviewer 1

Summary

The paper proposes a method for generating pairs of corresponding images in two different domains. The method does not rely on existence of paired images in the training set. The idea is to train a pair of GANs that share several layers of weights.

Qualitative Assessment

The paper is well written and a pleasure to read. The results are interesting and may be of use for the larger NIPS audience. I'm confused by the UDA section: specifically, how are the weights in f^(1)_2(.) trained? I don't find the cross-domain transformation results to be particularly convincing. Does this not rely heavily on a strong measure L of the difference between a pair of images, which is difficult to produce in practice? - Line 74: Use of 'top' and 'bottom' to describe layers is confusing, as there is no standard convention for the order of layers in a deep net. Especially given that the figures in the paper are horizontal. Best to use 'first' and 'last' instead. - Figure 3 caption: avergae -> average - Figure 3: What about performance when generative models share no layers? - Line 182: randon -> random - Figure 4 caption: pari -> pairs - Line 242: The learning hyperparameters was set -> The learning hyperparameters were set

Confidence in this Review

2-Confident (read it all; understood it all reasonably well)


Reviewer 2

Summary

The paper proposes a method for learning generative models of pairs of corresponding images belonging to two different domains (e.g. the RGB image of a scene and the corresponding depth image). The method is based on two adversarial networks with partially shared weights. The generative networks share the weights that map the noise to an intermediate code but have separate weights that map from the intermediate code to each image type. This induces the model to generate pairs of corresponding images even when it is never trained with corresponding image pairs. The authors evaluate the proposed method with several image datasets, and also provide a demonstration of how it can be applied to domain adaptation and cross-domain transformation problems.

Qualitative Assessment

I am in two minds about the paper. On the one hand the idea seems very interesting and powerful as it does not seem to rely on pairs of corresponding training images. The authors make a good effort of demonstrating and analyzing the capabilities of the approach in several ways (I especially like the unsupervised domain adaptation results). Visually the results look promising. (Although from the mostly qualitative evaluation it is not clear to what extent the models are overfitting to the training data, and e.g. for the RGB-D experiments I am finding it very hard to say anything more than that the generated pairs look superficially plausible.) But on the other hand, it seems to me that the task is not well defined. Since the the model is never presented with corresponding image pairs there is actually nothing in the training data that establishes what “corresponding” means. The only pressure for the network to establish a sensible correspondence between images in the two domains comes from the particular weight sharing constraint which allows each network only limited capacity to map from the shared intermediate layer to the two different types of images (the evaluation in the paper uses networks that use only one or two non-shared layers). This may be appropriate, and work well, for domain pairs that differ mostly in terms of low-level features (e.g. faces with blonde / non-blonde hair, or RGB and D images, as in the paper). But it makes me wonder how easy it would be to impose just the appropriate capacity constraint for domain pairs where the correspondence is at a more abstract level and/or more stochastic (e.g. images and text).

Confidence in this Review

1-Less confident (might not have understood significant parts)


Reviewer 3

Summary

The paper presents the coupled GAN (CoGAN) for learning to generate pair of images with different attributes or from different domains without knowing per-example correspondences. The key idea is to train a coupled GAN model jointly while sharing weights of the higher layers of both generative and discriminative models of GAN. In experiments, the paper demonstrates effectiveness of CoGAN at generating images at different domains and with different attributes. As an application of CoGAN, the paper proposes unsupervised domain adaptation as well as cross-domain image transformation, which shows promising results.

Qualitative Assessment

Strength: The proposed CoGAN is new and the experimental validation of the model is compelling. The generated pair of images of digits and faces are consistent while changing corresponding attributes. The performance of unsupervised domain adaptation, though it looks preliminary, is impressive. Weakness: The experiments with CoGAN on RGBD dataset is interesting but results are less convincing; for example, in the second example of Figure 5 bottom, the generated depth image seems to be correlated to the pixel values of RGB data (e.g., white wall is shown to be farther in the depth image but doesn't seem to be behind of the brown wall next to it). It will be great if the authors can provide some qualitative or quantitative results on the depth image prediction results following the procedure described in Section 5. Cross-domain image transformation (e.g., find z* for RGB image and generate corresponding depth image to compare against ground truth). It seems to me that the effectiveness of CoGAN in generating pair of images (or examples) is limited to pairs that are differ locally in the pixel space (e.g., edge of digits, parity of pixel values, hair color, ...). For example, can we generate pair of face images that are differ in view point (pose) with CoGAN? In line 120: [The CoGAN ... purely unsupervised fashion] is incorrect since CoGAN requires attribute labels for training.

Confidence in this Review

3-Expert (read the paper in detail, know the area, quite certain of my opinion)


Reviewer 4

Summary

This paper proposed a simple yet effective way to learn common semantics from data that have different low-level feature statistics. Fo example, for digits and their intensity inverted versions, the semantics are exactly the same while the pixel values are highly different. The paper achieves this by having two GANs with the high-level layers, which loosely correspond to semantics rather than feature encoding/decoding, shared between the two GANs. Training are then carried out in a purely unsupervised way without the need of explicit pairs of samples.

Qualitative Assessment

In general I think this is an elegant paper that learns high-level semantic correspondences in a purely unsupervised way. A key advantage in the proposed method is that, no explicit correspondence is needed to learn the semantics, and the correspondence emerges by simply analyzing the statistics of two separate datasets. Loosely speaking, the low-level feature descriptions are abstracted away by the unshared layers, while the high-level layers, being shared between the two GANs, learns the joint semantics of the two set of data distributions. One interesting question not yet answered by the paper is, to what level such corresponding semantics can be learned. The two experiments shown in the paper, MNIST and face generation, are relatively limited: MNIST examples show very simple distribution difference (edge vs non-edge, color inversion) that can be plausibly abstracted away via one layer neural net, and the face generation problem, strictly speaking, is not a CoGAN problem: there is only one underlying feature description. It would be great to have ablation study to show the limitation of the proposed method, and when/where things start to fail.

Confidence in this Review

2-Confident (read it all; understood it all reasonably well)


Reviewer 5

Summary

The paper proposes a framework for modeling pairs of corresponding images in two domains by combining two generative adversarial nets (GAN) and enforcing a simple weight-sharing constraint to these two nets. Experimental evaluations are carried out for pair image generation, domain adaptation and cross-domain image generation.

Qualitative Assessment

This paper generalizes the original generative adversarial net (GAN) to modeling pairs of corresponding images in two different domain by simply combining two GANs, each generating image in one domain. The simple weight-sharing strategy is used to capture the correspondence in the two domains. The proposed framework seems to work well. In the experimental evaluation, the learned models can generate reasonable pairs of corresponding images, especially they are useful in unsupervised domain adaptation. However, given the facts that the generative adversarial net (GAN) has proven to be a good model to generate images from one domain, and weight sharing is also a commonly used strategy with other deep nets for domain adaptation, the novelty in this paper is moderate. Also, the proposed framework includes 4 nets (each GAN has two nets) with 4 sets of weights, which might make the learning complicated and the interpretation difficult.

Confidence in this Review

2-Confident (read it all; understood it all reasonably well)


Reviewer 6

Summary

This paper proposes a very simple extension for generating images in two different domains. The model uses two GANs for each domain and shares the parameters for the lower level features in both generator and the discriminator. The authors supported their claims with experiments on MNIST, celebrity and RGBD datasets.

Qualitative Assessment

The paper proposes a very simple extension on GANs for multi-domain generation. The only novelty is the weight-sharing mechanism between two GANs. From the introduction, the motivation of this paper is not clear. It is not clear from the paper, in which applications or domains, generating pairs of images from different domains would be useful. It would be interesting to compare the proposed algorithm against the distillation as in done in [1] or Fit-Nets(or Fit-GANs)[2]. Authors should cite and may be compare against [3] or [4] as well. Since these papers are very related to the method proposed in this paper. There is a typo in the abstract: GoGAN --> CoGAN [1] Rusu, Andrei A., et al. "Policy distillation." arXiv preprint arXiv:1511.06295 (2015). [2] Romero, Adriana, et al. "Fitnets: Hints for thin deep nets." arXiv preprint arXiv:1412.6550 (2014). [3] Ganin, Yaroslav, and Victor Lempitsky. "Unsupervised domain adaptation by backpropagation." ICML 2015. [4] Ganin, Yaroslav, et al. "Domain-adversarial training of neural networks." JMLR (2016).

Confidence in this Review

2-Confident (read it all; understood it all reasonably well)